# Macrophage and Lymphocyte Infiltration Is Associated with Volumetric Tumor Size but Not with Volumetric Growth in the Tübingen Schwannoma Cohort

**DOI:** 10.3390/cancers13030466

**Published:** 2021-01-26

**Authors:** Vítor Moura Gonçalves, Elisa-Maria Suhm, Vanessa Ries, Marco Skardelly, Ghazaleh Tabatabai, Marcos Tatagiba, Jens Schittenhelm, Felix Behling

**Affiliations:** 1Department of Neurosurgery, University Hospital Tübingen, Eberhard-Karls-University Tübingen, 72076 Tübingen, Germany; vg81@sapo.pt (V.M.G.); elisa-maria.suhm@gmx.de (E.-M.S.); vanessa.ries92@web.de (V.R.); marco.skardelly@med.uni-tuebingen.de (M.S.); ghazaleh.tabatabai@uni-tuebingen.de (G.T.); marcos.tatagiba@med.uni-tuebingen.de (M.T.); 2Faculty of Medicine, University of Porto, 4200-319 Porto, Portugal; 3Center for Neuro-Oncology, Comprehensive Cancer Center Tübingen -Stuttgart, University Hospital Tübingen, Eberhard-Karls-University Tübingen, 72076 Tübingen, Germany; jens.schittenhelm@med.uni-tuebingen.de; 4Department of Neurology and Interdisciplinary Neuro-Oncology, University Hospital Tübingen, Eberhard-Karls-University Tübingen, 72076 Tübingen, Germany; 5Hertie Institute for Clinical Brain Research, 72076 Tübingen, Germany; 6German Cancer Consortium (DKTK), DKFZ Partner Site Tübingen, 72076 Tübingen, Germany; 7Department of Neuropathology, Institute of Pathology and Neuropathology, University Hospital Tübingen, Eberhard-Karls-University Tübingen, 72076 Tübingen, Germany

**Keywords:** vestibular schwannoma, acoustic neuroma, inflammation, macrophage, lymphocyte, immunohistochemistry, tumor volume, volumetric growth

## Abstract

**Simple Summary:**

Vestibular schwannomas are benign tumors arising from the 8th cranial nerve. With microsurgical resection or radiation therapy, most patients can be cured. However, recurrent tumors are difficult to treat, and there are no other established treatment options besides local treatment. Growing evidence of the oncogenic role of inflammatory processes in schwannomas gives hope to find treatable targets for innovative therapeutic strategies. To further define inflammatory cell infiltration in this tumor type, we analyzed tumor tissue for macrophage and lymphocyte infiltrates and compared it with volumetric tumor size and growth. Increased inflammatory cell infiltration was found to be associated with larger tumor size but not with volumetric growth.

**Abstract:**

Most patients with vestibular schwannomas can be cured with microsurgical resection, or tumor growth can be stabilized by radiotherapy in certain cases. Recurrence is rare but usually difficult to treat. Treatment alternatives to local therapies are not established. There is growing evidence of the role of inflammatory processes in schwannomas, which may be exploitable by targeted innovative therapies. To further define the impact of inflammation with tumor growth in vestibular schwannoma, we performed immunohistochemical analyses of CD3, CD8, CD68 and CD163 to assess lymphocyte and macrophage infiltration in 923 tumor tissue samples of surgically resected vestibular schwannomas. An inflammatory score was compared with tumor size and volumetric growth. We observed a significantly larger preoperative tumor size with increased expression rates of CD3, CD8, CD68 and CD163 (*p* < 0.0001, *p* < 0.0001, *p* = 0.0015 and *p* < 0.0001, respectively) but only a significant difference in percentual volumetric tumor growth for CD163 when regarding a CART-specified cut off. When all four markers were combined as an inflammatory score, there was no difference in percentual tumor growth. We conclude that inflammatory cell infiltration increases with larger tumor size but is not associated with percentual volumetric tumor growth.

## 1. Introduction

Vestibular schwannomas (VS) are benign, slow-growing tumors arising from the vestibular portion of the vestibulocochlear nerve. Due to its proximity to adjacent neurovascular structures, localized therapy via microsurgical resection or radiotherapy is necessary for growing or sizable lesions [1]. However, recurrent tumors are difficult to treat and especially patients suffering from neurofibromatosis type 2 (NF2) usually present with bilateral vestibular schwannomas at a young age and numerous additional neoplasms of the nervous system [2]. As an alternative treatment option, bevacizumab has been studied in NF2 patients with unconvincing results [3,4]. Other treatment options for vestibular schwannomas have not yet been established.

Within the last years, the role of inflammatory processes in the development and progression of vestibular schwannoma has been described [5,6,7]. Especially the infiltration of vs. tissue with inflammatory cells has been reported to be associated with tumor growth in small cohorts [8,9]. First insights into the role of inflammatory processes have been gained. For example, the differentiation of infiltrating macrophages into M2 macrophages has been described [6,10]. These cells are believed to act as tumor-associated macrophages, a concept that has been developed in other cancer types. It is suggested that the tumor microenvironment is able to recruit macrophages into functioning as supportive helpers regarding tumor growth [11]. With the rising significance of immunotherapeutic approaches in cancer and the first insights into vestibular schwannomas, the role of inflammation in vestibular schwannoma growth needs to be investigated in greater detail with higher case numbers. We have previously shown that the inflammatory mediator cyclooxygenase 2 is associated with tumor extension and proliferative marker expression [12]. This current retrospective study was designed to further investigate the role of immune cell infiltration by lymphocytes and macrophages and their correlation with tumor growth in a large cohort of vestibular schwannomas.

## 2. Results

### 2.1. Distribution of Immunohistochemical Marker Expression

The distribution of immunohistochemical expression of the markers CD3, CD8, CD68 and CD163 in 923 primary sporadic vestibular schwannomas is delineated in Figure 1. A difference in expression distribution was observed for lymphocyte and macrophage markers. For the lymphocyte markers CD3 and CD8, little to no expression (score 0) was observed in 162 (17.6%) and 118 (12.8%) cases, respectively. The majority of tumors showed a low immunopositive cell count (score 1: 523/919 (56.9%) and 544/920 (59.1%), respectively) while both markers showed a decreasing frequency for higher cell counts (distribution of scores 2, 3 and 4 are shown in Appendix A).

For the macrophage marker CD68, our analysis showed that 179 tumors had a score of 0 (19.5%), while the majority of cases reached a score of 1 (282/916 (30.8%)). Higher CD68 scores showed lesser frequencies. Little to no immunopositivity for CD163 (score 0) was observed for the majority of tumors (386/915 (42.2%)). A low cell count (score 1) was reached by 325/915 tumors (35.5%), and higher CD163 scores were seen in fewer cases (Figure 1D), similar to the distribution of CD68 scores (Figure 1C). Proliferative activity of tumors was measured with MIB1 expression with showed a mean value of 1.33% ranging from 0 to 4.75%.

### 2.2. Preoperative Volumetric Tumor Size and Correlation to Immunohistochemistry

Volumetric measurement of preoperative tumor volume was available for 767 cases (83.1%). The mean tumor volume was 4.73 cm^3^ ranging from 0.04 to 52.14 cm^3^.

Classification and regression tree analysis determined 1.57% MIB1 immunopositivity as the optimal cutoff in this cohort. Tumors with a MIB1 expression below 1.57% had a slightly larger preoperative tumor size compared to tumors with a higher MIB1 expression exceeding or equal 1.57%, but statistical significance was missed (4.93 compared to 4.03 cm^3^, *p* = 0.0770, see Figure 2A).

Significant differences in the mean preoperative tumor volume were also observed for all immune cell markers (CD3, CD8, CD68 and CD163). In contrast to MIB1, a higher preoperative tumor volume was seen for higher expression scores for all lymphocyte and macrophage markers, regarding the overall score. For the CART-derived cutoffs this was also the case except for CD3 (Figure 3 and Table 1).

### 2.3. Volumetric Tumor Growth

Volumetric tumor growth was assessed for 189 cases (20.5%). When regarding tumor growth as the difference in volume in cm^3^ per year, it showed similar significant differences across the inflammatory marker scores as the preoperative tumor volume. Thus, a higher expression score of CD3, CD8, CD68 and CD163 was associated with larger preoperative tumor size and faster volumetric growth in vestibular schwannomas (Appendix A). In contrast, the proliferative activity (MIB1 expression) did not show a significant difference in volumetric tumor growth in cm^3^ per year (Figure 2B). We next calculated the percentual volumetric tumor growth as a more representative marker for growth dynamic independent of the initial preoperative tumor volume. A MIB1 expression exceeding the cutoff at 1.57% had a significantly faster percentual volumetric tumor growth when compared to tumors with lower MIB1 expression (130.00 and 84.15%/year, respectively, *p* = 0.0032) (Figure 2C).

VS with a CD163 score of 0 showed a significant slower percentual tumor growth compared to tumors reaching a score of 1–4 (79.58 compared to 106.90%/year, respectively, *p* = 0.0465). Otherwise, the growth rate showed no difference when evaluating the immunohistochemical markers for lymphocyte (CD3 and CD8) and macrophage infiltration (CD68 and CD163). Results were not significant for the complete score and CART-specified cutoffs for each marker (Figure 4).

### 2.4. Inflammatory Score (IS)

For overall assessment an inflammatory score (IS) was generated from available immune cell marker data. An IS of 0 was given to 343 tumors (37.4%), meaning neither the CART-specified cut off for prominent lymphocytic nor macrophage infiltration was reached. A total of 535 cases (58.4%) received a score of 1 and 38 schwannomas (4.1%) reached the maximum score of 2, indicating prominent expression of lymphocyte and macrophage markers. 

The preoperative tumor volume was increased for tumors with a higher IS (Figure 5A). When both cohorts with scores 1 and 2 were combined the mean preoperative tumor volume reached 6.22 cm^3^, compared to 3.62 cm^3^ for tumors with a score of 0 (*p* < 0.0001, Figure 5B). In contrast to this, the percentual volumetric tumor growth was increased with higher inflammatory scores but without statistical significance (Figure 5C). 

### 2.5. Multivariate Analysis of Tumor Growth

To assess the impact of the proliferative marker MIB1 and the inflammatory score, a multivariate linear regression was used, and the previously established CART-specified cut offs were applied (Table 2). A MIB1 expression exceeding 1.57% was revealed as an independent factor for faster tumor growth *p* = 0.0103). On the contrary, the inflammatory score did not reach statistical significance.

## 3. Discussion

Our study provides insight into the correlation of immune cell infiltration and volumetric tumor growth in surgically resected primary vestibular schwannomas. The role of inflammation in the development and progression of schwannomas has gained increasing interest over the last years [6,7]. In 2000 Labit-Bouvier and colleagues assessed the expression of CD34 and CD45 in 69 vestibular schwannoma tumor samples. They described a significant association between the degree of inflammatory infiltrates and duration of symptoms, but not with tumor growth [13]. It must be noted that CD34 and CD45 are markers for hematopoietic cells in general and do not allow for differentiation according to immune cell type [14,15] and that tumor growth was not measured volumetrically.

A more specific macrophage marker analysis together with volumetric measurements was done by De Vries et al. in 2013 and 2019. The authors investigated the role of tumor associated macrophages in vestibular schwannomas by analyzing the expression of CD163, M-CSF and IL-34 in ten fast- and slow-growing vestibular schwannomas. A higher expression of M-CSF was seen in fast-growing VS and also in tumors with high CD163 expression [8,16]. The data suggests a role of macrophage infiltration in tumor growth. However, the case number is too small to draw solid conclusions. Although, the authors applied volumetric growth analysis, no percentual growth calculation was done. We learned from our large dataset, that the growth of volume per year, which de Vries et al. have applied, is not accurate since larger tumors show a bigger increase in volume. We therefore used the relative increase (per cent) in tumor volume per year which showed a significant association with the expression of the proliferation marker MIB1. With this approach a large increase of a small tumor and a little increase of a large tumor can be differentiated by its biological significance. Our data also showed that macrophage and lymphocyte marker expression was increased in larger tumors. For CD163 a significant difference was only reached when a CART-specified cut off was applied. A faster growth rate was associated with higher CD163 expression scores. This is in line with prior studies that suggested an impact of tumor-associated macrophages with VS growth. However, no impact of infiltration with lymphocyte markers or CD68 on tumor growth was observed in our cohort. Furthermore, when the infiltration of lymphocyte and macrophage marker expression was taken together as an inflammatory score, no difference in tumor growth was seen. This is in direct contradiction to the current literature that suggests that immune cell infiltration leads to faster tumor growth [6,8,17]. However, our study is the largest so far in regard to the description of immune cell infiltration and volumetric growth. Additionally, we applied a more detailed method of calculating volumetric growth in vestibular schwannoma. We also included quantification of MIB1 expression into a multivariate analysis which is an established factor for VS recurrence or regrowth [18,19]. Similar observations for MIB-1 correlations were made in a study with 7 growing VS [9]. Therefore, we believe that the findings of our study are valid. 

Based on the current literature, it has been advocated, that immune cell infiltration might be an expression of tumor associated macrophages driving schwannoma growth [6,8,17]. Only one study that assessed PET imaging with specific tracers for inflammation in a small group of vestibular schwannomas showed a higher cell density in non-growing tumors compared to growing VS [9]. Our results partially confirm this. CD163 expression is associated with tumor growth when dichotomized according to a CART-specific cut off. However, when looking at the complete score, there was no association with tumor growth. The immunohistochemical markers CD68 and CD163 for the detection of tumor-associated macrophages are widely used [20,21]. However, the quantification of immunohistochemistry of macrophage markers is not standardized and divergent results can be explained by technical limitations and antibody employment [22]. Our observed CD68 and CD163 staining characteristics fits well with the morphology of macrophages. We therefore believe that our expression analyses are valid and represent true immune cell infiltration in vestibular schwannomas. 

There are several explanations for the increase in immune cell infiltration of larger tumors. It is possible that the observed immune cell infiltration may be an expression of tissue changes due to angiogenesis, which may be increased with progressive tumor growth [23]

Furthermore, the presence of Antoni B areas is more common in vestibular schwannomas and is usually associated with the infiltration of macrophages and T lymphocytes. It is believed that Antoni B tissue is the result of the degeneration of Antoni A tissue [7,24], which would be more pronounced in larger/older tumors. This may explain our observation that tumors with higher lymphocyte and macrophage marker expression have a larger preoperative tumor size.

There is also growing evidence suggesting that schwannomas emerge after peripheral nerve trauma and subsequent faulty nerve regeneration processes. In vestibular schwannoma, this is believed to be due to noise exposure [7,25]. The immune cell infiltration associated with the regeneration process may be in line with the increased immune cell infiltration seen in larger tumors.

A subset of macrophages is characterized by the expression of proinflammatory cytokines such as IL-12, IL-1, and IL6. While IL-1beta and IL-6 are increased in schwannoma and may contribute to tumor growth [26], especially IL-12 exerts a robust antitumor response [27]. Future studies should therefore address potential shifts in IL composition in vs. during volumetric growth. A more detailed differentiation of macrophages and inflammatory processes is necessary for future studies. Especially markers to further describe the possible presence of tumor-associated macrophages or relevant target structures for immune checkpoint inhibition need to be considered. A few studies have given the first promising insights in small and selective cohorts [6,10].

It should also be noted that the growth dynamics in vs. are possibly more complex and unique than assumed. A vestibular schwannoma encounters surrounding structures of different resistance during its growth. In the beginning, it grows along the longitudinal axis of the internal acoustic canal, which is formed by the petrous bone. Some vs. even cause a widening of the internal acoustic canal. With further growth, the tumor encounters the mostly spacious cerebellopontine cistern, where it encounters little resistance until it reaches the cerebellar peduncle and brainstem. It can be hypothesized that the growth dynamic in tumor tissue changes depending on the size and extension of a vestibular schwannoma. More detailed volumetric growth analyses are necessary to address this question.

### Limitations

The main limitation is the retrospective nature of the study. Furthermore, the cohort is purely surgical since resected tumor tissue was assessed, and schwannomas treated conservatively or with radiotherapy only are not included. The volumetric measurements depended on the availability of sufficient preoperative imaging. As an international center for skull base surgery, many cases are referred to us with a clear indication for surgery due to the tumor size or symptomatic burden. Therefore, sequential preoperative imaging was only available for a subgroup of the study cohort. It is possible that especially vestibular schwannomas that present with large initial size are underrepresented in this subgroup analysis because observation is rarely chosen in such cases.

The tissue microarray method does not allow for immunohistochemical assessment of the whole tumor tissue. However, for each case, two tissue cylinders measuring 1 mm in diameter each were extracted from different representative areas after a detailed assessment of HE stains to address potential intertumor heterogeneity.

## 4. Materials and Methods

### 4.1. Patient Cohort

Between 10/2003 and 03/2017, 1143 vestibular schwannomas were surgically treated at the University Hospital Tübingen and were screened for inclusion. Tumors of NF2 patients and recurrent schwannomas were excluded as well as cases with missing consent for tissue analysis, incomplete clinical data or insufficient tissue for further processing. Overall, 923 primary vestibular schwannomas were included (Figure 6). Clinical data were retrieved from electronic patient files.

### 4.2. Volumetry

Data from preoperative magnetic resonance imaging (MRI) was reviewed, and volumetric analysis was performed using the planning software Brainlab iPlan Net, version 2.4 (Brainlab AG, Munich, Germany). Measurements were determined after selecting the most pertinent image sequence (either the post gadolinium T1-weighted or the T2 weighted images) and employing a semi-automatic image segmentation tool. Data from preoperative magnetic resonance imaging (MRI) with slice thickness ranging from 0.3 to 2.5 mm were used. For some cases, computer tomography (CT) data was accepted as well if postcontrast sequences with appropriate slice thickness were available. For the preoperative tumor volume measurement, images were only used if they were done six months or less prior to surgical resection. Volumetric tumor growth was calculated if at least two preoperative images with appropriate data quality and an interval of at least three months were available. All other cases were excluded from the volumetric assessment (Figure flow chart). Volumetric growth was calculated as the percentual growth based on the volumetric size of the tumor. To define cutoffs for each marker with the most pronounced difference in percentual volumetric tumor growth classification and regression tree analyses (CART) were performed for MIB1, CD3, CD8, CD68 and CD163. One extreme outlier with a percentual volumetric tumor growth rate of 5192.1%/year was excluded. For comparison, the second highest growth rate was 385.7%/year.

### 4.3. Tissue Microarray Construction and Immunohistochemistry

For the construction of tissue microarrays (TMA), 1 mm tissue cylinders were extracted from formalin-fixed and paraffin-embedded tumor samples archived in the department Neuropathology after evaluation and marking of the respective hematoxylin and eosin stains. For most of the cases, enough tissue was available to retrieve two tissue probes. A conventional microarrayer was used (Beecher Instruments, Sun Prairie, WI, USA). TMAs were cut with a microtome, and 4 μm tissue slices were produced and dried at 80° for 15 min. Immunohistochemical staining was done with a Ventana BenchMark immunostainer (Ventana Medical Systems, Tucson, AZ, USA).

Pretreatment with cell conditioning solution CC1 (pH 8.5) was done for 14 (CD68), 40 (MIB1, CD3 and CD163) or 64 min (CD8) followed by primary antibody incubation at 37° (CD8 (ready to use, Roche, Basel, Switzerland) and CD163 (1:1000, ABD Serotec, Puchheim, Germany)) or 42 °C (MIB1 (1:200, DAKO, Santa Clara, CA, USA), CD3 (1:500, Thermo Fisher Scientific, Waltham, MA, USA) and CD68 (1:200, Agilent DAKO, Santa Clara, CA, USA)). Subsequently, OptiView HQ universal linker was applied for 12 min, followed by incubation with OptiView HRP Multimer for 12 min. Counterstaining was done with hematoxylin for 4 min.

As controls, the human cerebral and cerebellar cortex, as well as a sample of a colorectal carcinoma metastasis, were placed on each TMA block.

### 4.4. Microscopic Assessment and Inflammatory Score (IS)

The expression of CD3 and CD8 was determined by manual counting of stained cells of the complete 1 mm biopsy punch. If the tissue cylinder on the tissue microarray was incomplete, the result was multiplied with the completeness of the assessed area (e.g., count of immunopositive cells x 0.5 for a 50% tissue cylinder). The results were arranged in a semiquantitative score according to the immunopositive cell count per 1 mm tumor tissue area. Quantification of CD68 and CD163 was done with a semiquantitative score by estimation of percentual immunopositivity (Table 3). Examples for low and high expression of all lymphocyte and macrophage marker stainings are displayed in Figure 7.

In order to combine all assessed immune cell markers, an inflammatory score (IS) was designed. Tumors received one point for lymphocytic infiltration, if the immunohistochemical result for CD3 or CD8 exceeded the CART-specified cut offs (cut off score 4 for both markers). A second point was given for macrophage infiltration if the immunohistochemical result for CD68 and CD163 reached the CART-specified cut offs (score > 2 and >0, respectively). 

MIB1 expression was analyzed with the help of an automated percentual assessment of digital images taken from stained full tumor slides. Image J software (Version 1.51j8, NIH, Bethesda, MD, USA) together with the plugins Bio-Formats (Release 5.4.1; Open Microscopy Environment, Madison, NJ, USA) and ImmunoRatio (Version 1.0c, Institute of Biomedical Technology, University of Tampere, Tampere, Finland) were used (Appendix A).

### 4.5. Statistical Methods

Statistical analysis was done with JMP^®^ Statistical Discovery Software, version 15.1.0 (Cary, NC, USA: SAS Institute Inc.; 1989). An analysis of variance was done as well as multivariate linear logistic regression analysis. A significance level of α < 0.05 was applied. 

## 5. Conclusions

Increased infiltration of vestibular schwannomas with immune cells is associated with larger tumor size, but not with faster tumor growth.

## Figures and Tables

**Figure 1 cancers-13-00466-f001:**
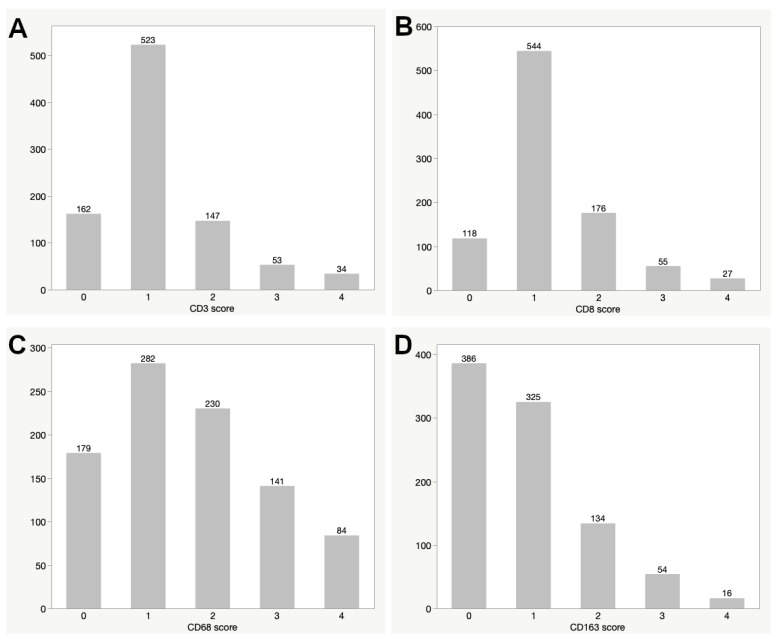
Distribution of the immunohistochemical expression score for CD3 (**A**), CD8 (**B**), CD68 (**C**) and CD163 (**D**).

**Figure 2 cancers-13-00466-f002:**
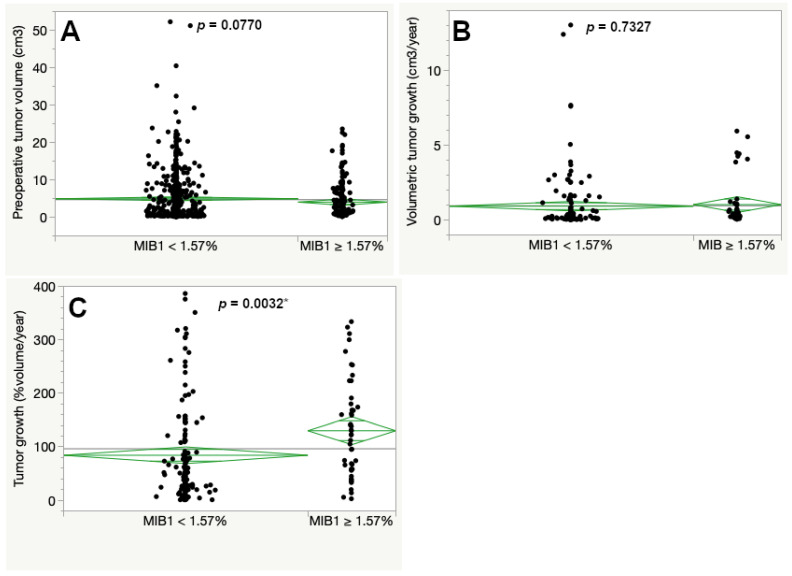
Preoperative tumor volume (**A**), volumetric tumor growth (**B**) and percentual volumetric (**C**). The asterisk (*) marks statistically significant results. Table 1 expression in the tumor tissue. The cutoff at 1.57% was set according to a classification and regression tree (CART) analysis regarding percentual volumetric growth (ANOVA, asterisk (*) marks statistically significant results).

**Figure 3 cancers-13-00466-f003:**
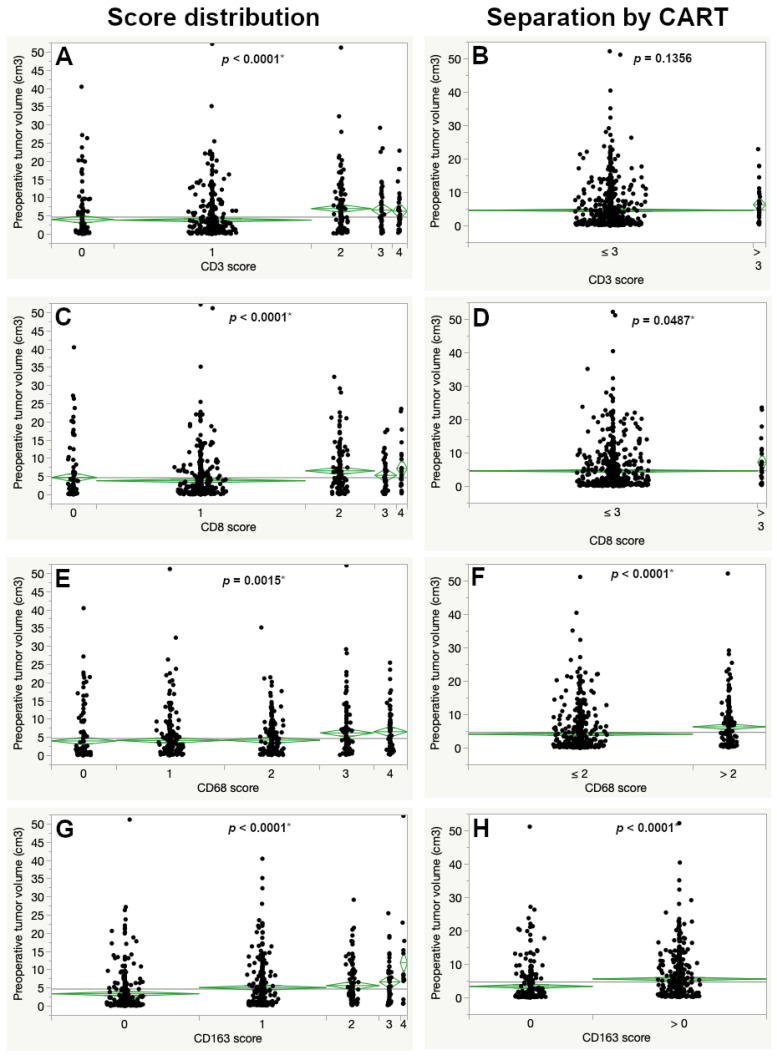
Differences in preoperative tumor volume according to the immunohistochemical expression across the complete immunohistochemistry score (left images) and after determining the CART-specific cutoff (right images) for CD3 (**A**,**B**), CD8 (**C**,**D**), CD68 (**E**,**F**) and CD163 (**G**,**H**). Significantly larger preoperative tumor volumes were seen with increased expression of each marker (ANOVA). The asterisk (*) marks statistically significant results.

**Figure 4 cancers-13-00466-f004:**
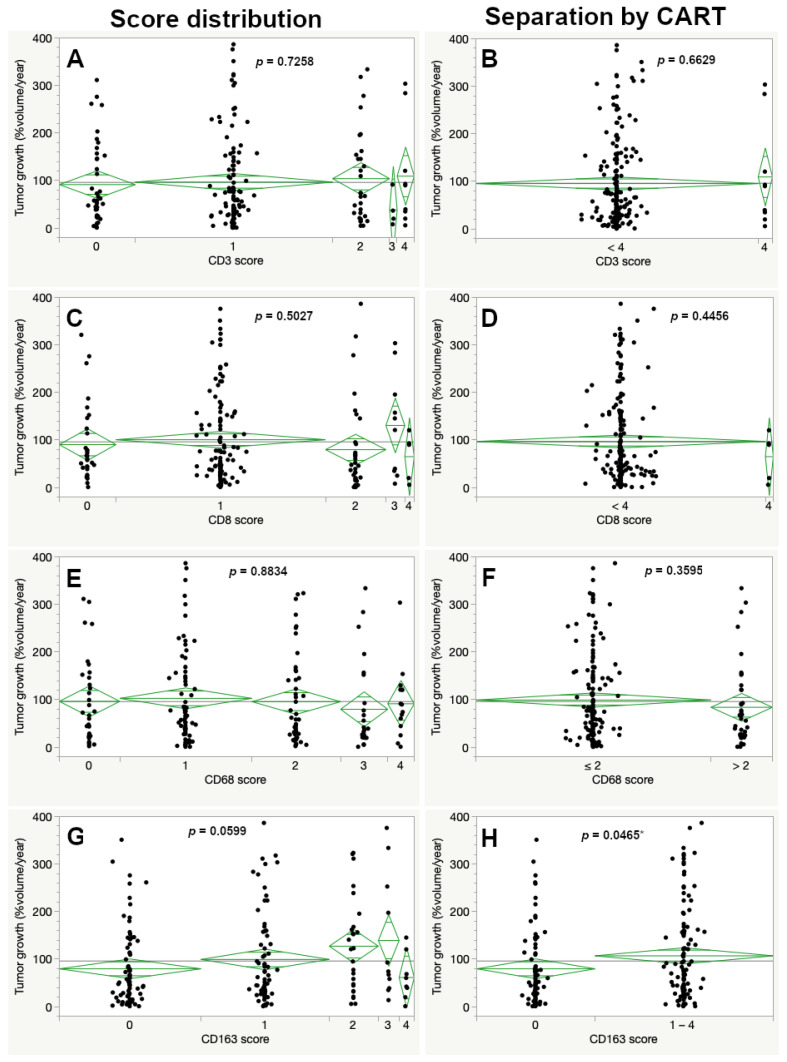
Percentual volumetric tumor growth according to the immunohistochemical expression for the complete immunohistochemistry score (left images) and the CART-specific cutoff (right images) for CD3 (**A**,**B**), CD8 (**C**,**D**), CD68 (**E**,**F**) and CD163 (**G**,**H**). No significant differences in tumor growth were seen, except for the CART-specific cut off for CD163 (ANOVA, asterisk (*) marks statistically significant results).

**Figure 5 cancers-13-00466-f005:**
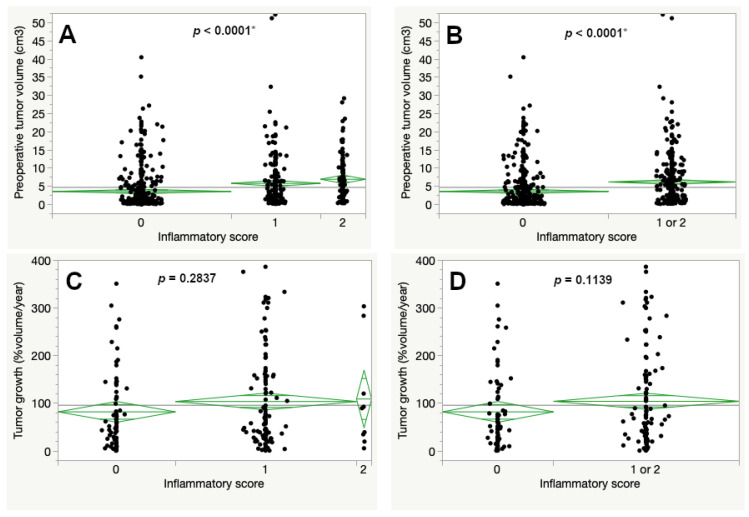
Preoperative tumor volume (**A**,**B**) and percentual volumetric tumor growth (**C**,**D**) according to the inflammatory score (ANOVA, asterisk (*) marks statistically significant results).

**Figure 6 cancers-13-00466-f006:**
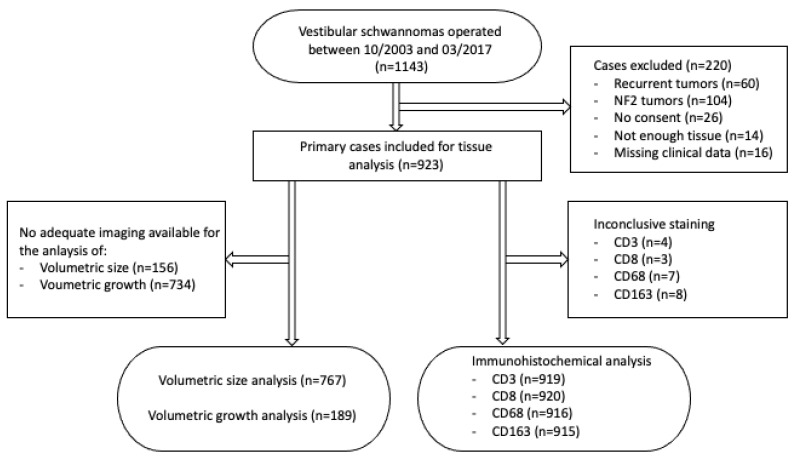
Consort diagram showing the cases included and excluded in this study.

**Figure 7 cancers-13-00466-f007:**
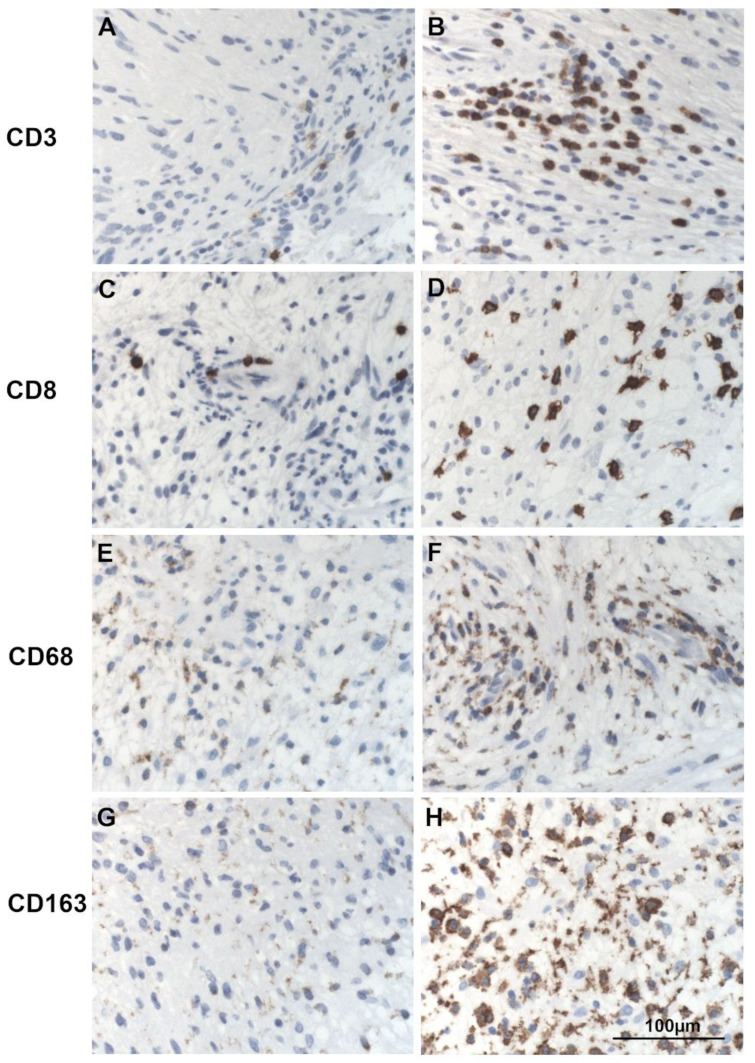
Examples of the immunohistochemical expression differences of lymphocytic and macrophage markers in vestibular schwannoma. Cytoplasmic staining of CD3 and CD8 is shown in tumor areas with little (**A**,**C**) and more pronounced expression (**B**,**D**). The often perinuclear pronounced cytoplasmatic staining of CD68 and CD163 can be seen for cases with low to absent (**E**,**G**) and high expression (**F**,**H**).

**Table 1 cancers-13-00466-t001:** Preoperative volumetry according to inflammatory cell marker expression.

Variable	*N* (%)	Mean Tumor Volume in cm^3^	*p*-Value (ANOVA)
CD3 score			
0	134 (17.5)	4.05	<0.0001 *
1	424 (55.5)	3.9	
2	129 (16.9)	7.06	
3	45 (5.9)	6.66	
4	32 (4.2)	6.34	
≤3	732 (95.8)	4.65	0.1356
>3	32 (4.2)	6.34	
CD8 score			
0	97 (12.7)	4.78	<0.0001 *
1	449 (58.7)	3.88	
2	149 (19.5)	6.6	
3	47 (6.1)	5.37	
4	23 (3.0)	7.24	
≤3	742 (97.0)	4.64	0.0487 *
>3	23 (3.0)	7.24	
CD68 score			
0	139 (18.2)	4.04	0.0015 *
1	231 (30.3)	4.19	
2	205 (26.9)	4.25	
3	115 (15.1)	6.23	
4	72 (9.5)	6.58	
≤2	575 (75.5)	4.18	<0.0001 *
>2	187 (24.5)	6.36	
CD163 score			
0	316 (41.5)	3.42	<0.0001 *
1	272 (35.7)	5.11	
2	113 (14.8)	5.67	
3	45 (6.0)	6.67	
4	15 (2.0)	11.97	
0	316 (41.5)	3.42	<0.0001 *
>0	445 (58.5)	5.64	
Inflammatory score			
0	435 (57.1)	3.62	<0.0001 *
1	217 (28.5)	5.84	
2	110 (14.4)	6.97	
0	435 (57.1)	3.62	<0.0001 *
1 or 2	327 (42.9)	6.22	

Asterisk (*) marks statistically significant results.

**Table 2 cancers-13-00466-t002:** Multivariate linear regression of percentual volumetric tumor growth.

	Estimate	Std Error	t Ratio	Lower 95%	Upper 95%	*p*-Value
Intercept	104.30	8.29	12.58	87.94	120.66	<0.0001 *
MIB1 >= 1.57%	−20.92	8.07	12.58	−36.83	−5.00	0.0103 *
Inflammatory score 0	−7.91	7.33	−1.08	−22.38	6.55	0.2818

Asterisk (*) marks statistically significant results.

**Table 3 cancers-13-00466-t003:** Description of the semiquantitative scoring system for lymphocyte and macrophage markers.

Variable	Score	0	1	2	3	4
CD3/8	Immunopositive cell count/1 mm	0–5	5–50	50–100	100–150	>150
CD68/163	Immunopositive area in %/1 mm	0–5	5–25	25–50	50–75	75–100

## Data Availability

The dataset is available from the corresponding author upon reasonable request.

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
