# Peer review of "Macrophage and Lymphocyte Infiltration Is Associated with Volumetric Tumor Size but Not with Volumetric Growth in the Tübingen Schwannoma Cohort"

_cancers, 2021, doi:10.3390/cancers13030466_

Round 1

Reviewer 1 Report

I really enjoyed Dr. Gonclaves and group's paper regarding a large group of patients (nearly 1000) whom underwent surgical resection and underwent immune analysis of the type of cells within the tumor.  They do a nice job in writing and analysis.  They describe a novel conclusion in the discussion "Based on the current literature, the observation of decreased tumor growth with increased inflammatory infiltration was unexpected." 

1.Currently, the paper fails to recognize the push and pull of inflamation present within these tumors and it should be recognized that some form of subtyping should be performed as demonstrated in this paper:

Predominance of M1 subtype among tumor-associated macrophages in phenotypically aggressive sporadic
vestibular schwannoma
Avital Perry et al. 

I would encourage the authors to perform a subtype analysis of the inflammatory infiltrate in order to further understand the complex immune environment.  

2.  The value of the "inflammatory score" is unknown as this number is a combination of random inflammatory cells that may have divisive affects on the tumor.  I would have a pathologist and immunologist weigh in on the valaditity of this data.  

3.  we would like to get data on tumor recurrence or progression after surgery relative to the cell analysis, if this is possible please include.  

Author Response

Response to Reviewer 1:

I really enjoyed Dr. Gonclaves and group's paper regarding a large group of patients (nearly 1000) whom underwent surgical resection and underwent immune analysis of the type of cells within the tumor.  They do a nice job in writing and analysis.  They describe a novel conclusion in the discussion "Based on the current literature, the observation of decreased tumor growth with increased inflammatory infiltration was unexpected." 

1.Currently, the paper fails to recognize the push and pull of inflamation present within these tumors and it should be recognized that some form of subtyping should be performed as demonstrated in this paper:

Predominance of M1 subtype among tumor-associated macrophages in phenotypically aggressive sporadic vestibular schwannoma (Avital Perry et al.) 

I would encourage the authors to perform a subtype analysis of the inflammatory infiltrate in order to further understand the complex immune environment.  

We thank the reviewer for this helpful comment. The role of inflammation seems to be more complex in vestibular schwannoma and we have added a paragraph in the discussion to express the need to further define the inflammatory processes in vestibular schwannomas in future studies. Currently our CD163 staining allows the identification of macrophages with M2 phenotype (Kwiecien et al. 2019, PMID: 32140052). We plan to perform a more detailed M1/M2 subtype analysis in our cohort in the future.

Page 11, line 254-258:

“A more detailed differentiation of macrophages and inflammatory processes is necessary in future studies. Especially markers to further describe the possible presence of tumor associated macrophages or immune checkpoint inhibition need to be considered. A few studies have given first promising insights in small and selective cohorts[6,10].”

  1. The value of the "inflammatory score" is unknown as this number is a combination of random inflammatory cells that may have divisive affects on the tumor.  I would have a pathologist and immunologist weigh in on the valaditity of this data.  

The inflammatory score was generated as a simple score to express the combined expression of all inflammatory markers that were assessed in this study. For each marker a cut off according to a CART analysis was done in order a have the most pronounced difference of the subgroups regarding radiographic tumor growth. The score was designed together with a board-certified neuropathologist.

  1. we would like to get data on tumor recurrence or progression after surgery relative to the cell analysis, if this is possible please include.  

So far, our data only allows conclusions on the preoperative tumor growth and resected tissue. We agree with the reviewer that data on tumor recurrence/progression is of great interest and we plan to gather this information for future expansion of our research project.

Reviewer 2 Report

REVIEW For Cancers 1056080

December 29, 2020

General Comments

 I think this is an interesting basic research study to investigate the impact of inflammation with tumor growth in vestibular schwannoma. Authors performed immunohistochemical analyses of CD3, CD8, CD68 and 41 CD163 to assess lymphocyte and macrophage infiltration in 923 tumor tissue samples of surgically resected vestibular schwannomas.  

Then authors concluded that a significantly larger preoperative tumor size with increased expression rates of CD3, CD8, CD68 and CD163 (p<0.0001, p<0.0001, p=0.0015 and p<0.0001, respectively) but not differences in percentual volumetric tumor growth. When all four markers four markers were combined as an inflammatory score, tumors with high inflammatory infiltration showed a slower percentual growth in a multivariate analysis including MIB1 expression (p=0.0249). Authors conclude that inflammatory cell infiltration increases with larger tumor size but is associated with slower percentual volumetric tumor growth.

However, the basic hypothesis in terms of tumor immunology and defense mechanism looks very poor and immature.

We hope the authors could provide some perspective in the tumor immunology for vestibular schwannoma could help a lot to understand this project.

Author Response

Response to Reviewer 2:

I think this is an interesting basic research study to investigate the impact of inflammation with tumor growth in vestibular schwannoma. Authors performed immunohistochemical analyses of CD3, CD8, CD68 and 41 CD163 to assess lymphocyte and macrophage infiltration in 923 tumor tissue samples of surgically resected vestibular schwannomas.  

Then authors concluded that a significantly larger preoperative tumor size with increased expression rates of CD3, CD8, CD68 and CD163 (p<0.0001, p<0.0001, p=0.0015 and p<0.0001, respectively) but not differences in percentual volumetric tumor growth. When all four markers four markers were combined as an inflammatory score, tumors with high inflammatory infiltration showed a slower percentual growth in a multivariate analysis including MIB1 expression (p=0.0249). Authors conclude that inflammatory cell infiltration increases with larger tumor size but is associated with slower percentual volumetric tumor growth.

However, the basic hypothesis in terms of tumor immunology and defense mechanism looks very poor and immature.

We hope the authors could provide some perspective in the tumor immunology for vestibular schwannoma could help a lot to understand this project.

We regret to have provided an incomplete immunologic basis of our hypothesis and understand that this needs to be improved. We have added the following paragraph in the introduction.

Page 2, line 76-81:

[8,9]. First insights into the role of inflammatory processes have been gained. For example, the differentiation of infiltrating macrophages into M2 macrophages has been described[6,10]. These cells are believed to act as tumor-associated macrophages, a concept that has been developed in other cancer types. It is suggested that the tumor microenvironment is able to recruit macrophages into functioning as supportive helpers regarding tumor growth[11].

Reviewer 3 Report

I read your article with great interest, as a neuro-oncologist treating many vestibular schwannomas (VSs) by gamma knife. I think that your study is remarkable because it is the largest in regard to the description of immune cell infiltration in VSs. Although there are some unexplained results, I hope that studies like yours would develop the targeted therapy for VS in the near future. Unfortunately, we cannot currently make use of such medicines. In EANO guideline in 2020, observation or stereotactic radiosurgery (SRS) is recommended for most VSs unless they cause brainstem compression. So, I disagree with the statement in line 57-58  " Most patients can be cured with microsurgical excision while small VS can effectively be irradiated". In fact, most VSs are treated with SRS without consulting surgeons.  

Author Response

Response to reviewer 3: 

I read your article with great interest, as a neuro-oncologist treating many vestibular schwannomas (VSs) by gamma knife. I think that your study is remarkable because it is the largest in regard to the description of immune cell infiltration in VSs. Although there are some unexplained results, I hope that studies like yours would develop the targeted therapy for VS in the near future. Unfortunately, we cannot currently make use of such medicines. In EANO guideline in 2020, observation or stereotactic radiosurgery (SRS) is recommended for most VSs unless they cause brainstem compression. So, I disagree with the statement in line 57-58  " Most patients can be cured with microsurgical excision while small VS can effectively be irradiated". In fact, most VSs are treated with SRS without consulting surgeons. 

We thank the reviewer for his comment and interest in our study. We agree that many small vestibular schwannomas can be successfully treated by SRS and that it is an established treatment option as we have expressed in our introduction. The EANO guideline of 2020 confirms that the level of evidence to provide recommendations for VS is low compared to other intracranial tumors, and that the choice of treatment depends on clinical presentation, tumor size and expertise of the treating center. Our protocol as an experienced center for the treatment of VS with a case load of > 120 surgeries/year, includes surgery in young patients and in large tumors, in order to achieve a high long-term tumor control rate. We agree that this protocol cannot be applied to each center, so that we have made the following changes in order to avoid misunderstandings. 

Page 2, line 67-69:

Old version:

“Due to its proximity to adjacent neurovascular structures localized therapy is necessary for growing or sizable lesions. Most patients can be cured with microsurgical excision while small VS can effectively be irradiated.”

New version:

“Due to its proximity to adjacent neurovascular structures localized therapy via microsurgical resection or radiotherapy is necessary for growing or sizable lesions.”

Round 2

Reviewer 1 Report

The authors did not address my concerns, they stated while they had tumor specimens for the 1000 patients they could not correlate this with recurrence or progression which should be readily available in a complete data set.  Also, with the confounding results of macrophage infiltrate within the context of the rest of current papers published on this topic we asked for macrophage subtyping and they stated this would be a future research topic but is pertinent to this paper.  This of the 3 requests would be the hardest to complete in a review period and I understand this.  Further, we asked that someone other than a surgeon weigh in on the value of the inflammatory score which does not seem to be pertinent or something that will be valuable to this subset in the future and they did not.  My concerns were not addressed, I would leave it up to the editor to decide if they should address these as I believe they will substantially improve the paper and should be readily available for a paper to be accepted to a quality journal like Cancer.  of course the publication can be published without addressing these but the data will be incomplete here.  

summary of edits and responses:

Response to Reviewer 1:

I really enjoyed Dr. Gonclaves and group's paper regarding a large group of patients (nearly 1000) whom underwent surgical resection and underwent immune analysis of the type of cells within the tumor.  They do a nice job in writing and analysis.  They describe a novel conclusion in the discussion "Based on the current literature, the observation of decreased tumor growth with increased inflammatory infiltration was unexpected." 

1.Currently, the paper fails to recognize the push and pull of inflamation present within these tumors and it should be recognized that some form of subtyping should be performed as demonstrated in this paper:

Predominance of M1 subtype among tumor-associated macrophages in phenotypically aggressive sporadic vestibular schwannoma (Avital Perry et al.) 

I would encourage the authors to perform a subtype analysis of the inflammatory infiltrate in order to further understand the complex immune environment.  

We thank the reviewer for this helpful comment. The role of inflammation seems to be more complex in vestibular schwannoma and we have added a paragraph in the discussion to express the need to further define the inflammatory processes in vestibular schwannomas in future studies. Currently our CD163 staining allows the identification of macrophages with M2 phenotype (Kwiecien et al. 2019, PMID: 32140052). We plan to perform a more detailed M1/M2 subtype analysis in our cohort in the future.

Page 11, line 254-258:

“A more detailed differentiation of macrophages and inflammatory processes is necessary in future studies. Especially markers to further describe the possible presence of tumor associated macrophages or immune checkpoint inhibition need to be considered. A few studies have given first promising insights in small and selective cohorts[6,10].”

  1. The value of the "inflammatory score" is unknown as this number is a combination of random inflammatory cells that may have divisive affects on the tumor.  I would have a pathologist and immunologist weigh in on the valaditity of this data.  

The inflammatory score was generated as a simple score to express the combined expression of all inflammatory markers that were assessed in this study. For each marker a cut off according to a CART analysis was done in order a have the most pronounced difference of the subgroups regarding radiographic tumor growth. The score was designed together with a board-certified neuropathologist.

  1. we would like to get data on tumor recurrence or progression after surgery relative to the cell analysis, if this is possible please include.  

So far, our data only allows conclusions on the preoperative tumor growth and resected tissue. We agree with the reviewer that data on tumor recurrence/progression is of great interest and we plan to gather this information for future expansion of our research project.

Author Response

The authors did not address my concerns, they stated while they had tumor specimens for the 1000 patients they could not correlate this with recurrence or progression which should be readily available in a complete data set. 

As we have mentioned before, we completely agree that the recurrence/progression data are of interest for future studies. However, the manuscript focuses on the preoperative tumor growth and not the tumor recurrence/progression and therefore the dataset is complete for the current study.

Also, with the confounding results of macrophage infiltrate within the context of the rest of current papers published on this topic we asked for macrophage subtyping and they stated this would be a future research topic but is pertinent to this paper.  This of the 3 requests would be the hardest to complete in a review period and I understand this. 

Further macrophage subtyping is of high interest but also not the focus of this study. Introducing a new marker into the analysis of this large cohort is not feasible in a short time period.

Further, we asked that someone other than a surgeon weigh in on the value of the inflammatory score which does not seem to be pertinent or something that will be valuable to this subset in the future and they did not. 

We have stated during the first round of revision, that we have constructed the study together with a neuropathologist, who also shared senior authorship. Therefore, we have to contradict the reviewer’s statement that we have not adequately answered this question in the first round of revision.

My concerns were not addressed, I would leave it up to the editor to decide if they should address these as I believe they will substantially improve the paper and should be readily available for a paper to be accepted to a quality journal like Cancer.  of course the publication can be published without addressing these but the data will be incomplete here.

Overall, we are very thankful for the important input by the reviewer. However, the far-reaching changes suggested by the reviewer are beyond the focus of the manuscript and are also not feasible in a short time period. Therefore, we have not made any further changes. o

summary of edits and responses:

Response to Reviewer 1:

I really enjoyed Dr. Gonclaves and group's paper regarding a large group of patients (nearly 1000) whom underwent surgical resection and underwent immune analysis of the type of cells within the tumor.  They do a nice job in writing and analysis.  They describe a novel conclusion in the discussion "Based on the current literature, the observation of decreased tumor growth with increased inflammatory infiltration was unexpected." 

1.Currently, the paper fails to recognize the push and pull of inflamation present within these tumors and it should be recognized that some form of subtyping should be performed as demonstrated in this paper:

Predominance of M1 subtype among tumor-associated macrophages in phenotypically aggressive sporadic vestibular schwannoma (Avital Perry et al.) 

I would encourage the authors to perform a subtype analysis of the inflammatory infiltrate in order to further understand the complex immune environment.  

We thank the reviewer for this helpful comment. The role of inflammation seems to be more complex in vestibular schwannoma and we have added a paragraph in the discussion to express the need to further define the inflammatory processes in vestibular schwannomas in future studies. Currently our CD163 staining allows the identification of macrophages with M2 phenotype (Kwiecien et al. 2019, PMID: 32140052). We plan to perform a more detailed M1/M2 subtype analysis in our cohort in the future.

Page 11, line 254-258:

“A more detailed differentiation of macrophages and inflammatory processes is necessary in future studies. Especially markers to further describe the possible presence of tumor associated macrophages or immune checkpoint inhibition need to be considered. A few studies have given first promising insights in small and selective cohorts[6,10].”

  1. The value of the "inflammatory score" is unknown as this number is a combination of random inflammatory cells that may have divisive affects on the tumor.  I would have a pathologist and immunologist weigh in on the valaditity of this data.  

The inflammatory score was generated as a simple score to express the combined expression of all inflammatory markers that were assessed in this study. For each marker a cut off according to a CART analysis was done in order a have the most pronounced difference of the subgroups regarding radiographic tumor growth. The score was designed together with a board-certified neuropathologist.

  1. we would like to get data on tumor recurrence or progression after surgery relative to the cell analysis, if this is possible please include.  

So far, our data only allows conclusions on the preoperative tumor growth and resected tissue. We agree with the reviewer that data on tumor recurrence/progression is of great interest and we plan to gather this information for future expansion of our research project.

Round 3

Reviewer 1 Report

The impact of the paper would be greatly enhanced by the data that was asked for.  The authors have stated they plan to collect this data for future papers, however correlated the current data with recurrence should be very easy.  Since in two reviews it was not provided as I stated in my last review the addition of that data would make this a more complete dataset.  It appears to be the editors decision if the paper is suitable for the journal in its present form.